# Diaphragm Ultrasound in Critically Ill Patients on Mechanical Ventilation—Evolving Concepts

**DOI:** 10.3390/diagnostics13061116

**Published:** 2023-03-15

**Authors:** Pauliane Vieira Santana, Letícia Zumpano Cardenas, Andre Luis Pereira de Albuquerque

**Affiliations:** 1Intensive Care Unit, AC Camargo Cancer Center, São Paulo 01509-011, Brazil; 2Intensive Care Unit, Physical Therapy Department, AC Camargo Cancer Center, São Paulo 01509-011, Brazil; 3Pulmonary Division, Faculdade de Medicina da Universidade de São Paulo, São Paulo 05403-000, Brazil; 4Sírio-Libanês Teaching and Research Institute, Hospital Sírio Libanês, São Paulo 01308-060, Brazil

**Keywords:** diaphragmatic ultrasound, diaphragm dysfunction, myotrauma, diaphragm-protective mechanical ventilation, patient–ventilator asynchrony, mechanical ventilation, inspiratory effort

## Abstract

Mechanical ventilation (MV) is a life-saving respiratory support therapy, but MV can lead to diaphragm muscle injury (myotrauma) and induce diaphragmatic dysfunction (DD). DD is relevant because it is highly prevalent and associated with significant adverse outcomes, including prolonged ventilation, weaning failures, and mortality. The main mechanisms involved in the occurrence of myotrauma are associated with inadequate MV support in adapting to the patient’s respiratory effort (over- and under-assistance) and as a result of patient-ventilator asynchrony (PVA). The recognition of these mechanisms associated with myotrauma forced the development of myotrauma prevention strategies (MV with diaphragm protection), mainly based on titration of appropriate levels of inspiratory effort (to avoid over- and under-assistance) and to avoid PVA. Protecting the diaphragm during MV therefore requires the use of tools to monitor diaphragmatic effort and detect PVA. Diaphragm ultrasound is a non-invasive technique that can be used to monitor diaphragm function, to assess PVA, and potentially help to define diaphragmatic effort with protective ventilation. This review aims to provide clinicians with an overview of the relevance of DD and the main mechanisms underlying myotrauma, as well as the most current strategies aimed at minimizing the occurrence of myotrauma with special emphasis on the role of ultrasound in monitoring diaphragm function.

## 1. Introduction

Mechanical ventilation (MV) aims to ensure adequate gas exchange and to totally or partially replace the action of the respiratory muscles, which can be overloaded during acute respiratory failure. However, MV has widely recognized potentially harmful pulmonary effects [1,2] leading to the proposition of several ventilation strategies to limit the mechanical stress applied to the lungs [3]. More recently, it has been identified that MV can also injure the diaphragm due to a variety of supposed mechanisms, referred to as “myotrauma” [4], which leads to the impairment of the diaphragm function, referred to as ventilator-induced diaphragmatic dysfunction—VIDD [5].

VIDD is conceptualized as a reduction in the diaphragmatic force-generating capacity, specifically related to the use of MV [5] that occurs shortly after the onset of the MV [6] in a progressive, time-dependent manner [7], and is influenced by different MV modes and associated risk factors [7].

The recognition of diaphragm dysfunction (DD) is relevant in critically ill patients because it is a common event that occurs in the different stages of critical illness, since admission [8], during the early stages of MV [7], during weaning [9,10,11], and even in the recovery stages [12]. In addition, critically ill patients are often vulnerable to numerous risk factors concomitant with MV, such as sepsis, organ dysfunction, and medications that may contribute to diaphragmatic injury [8,13,14,15]. Also, there is compelling evidence on the unfavorable impact of diaphragmatic dysfunction on morbidity in critically ill patients, including weaning difficulties [7,8,9,10,11], ICU readmission [16], and mortality [8,17].

Considering the relevance of diaphragmatic dysfunction in critically ill patients, the current literature has been committed to prioritizing the care of patients under MV in order to monitor diaphragm function in the ICU and to minimize or avoid diaphragm myotrauma to prevent the development of VIDD [18].

Knowledge about the main mechanisms involved in the occurrence of myotrauma underlies the strategies that can be used to protect the diaphragm during MV [19,20].

The purpose of this review is to explore evolving concepts about the use of diaphragm ultrasound as a technique not only to assess diaphragm function but also to help monitor diaphragmatic effort for protective ventilation. We present an overview of the relevance of diaphragm dysfunction in critically ill patients on MV, its main mechanisms, and strategies to prevent it, focusing on ultrasound as a tool for monitoring diaphragmatic function.

## 2. Diaphragm Myotrauma-Mechanisms

Evidence of diaphragmatic muscle injury in critically ill mechanically ventilated patients has been widely appreciated and recognized [21] and is based on several clinical [7,22,23,24] and experimental studies [25,26,27,28,29].

To understand diaphragmatic dysfunction in critically ill patients, efforts have been made to standardize terminology, with the aim of ensuring communication between researchers. Thus, the term *myotrauma* has been adopted to refer broadly to the ventilator-associated diaphragmatic acute muscle injury [4].

According to the most current evidence, myotrauma is presumably associated with several mechanisms linked to the inappropriate management of MV, mainly related to the inadequacy of ventilatory support to suit the patient’s respiratory effort (over- and under-assistance) and as a result of AVP. The evidence for these mechanisms is summarized below [4]:Over-assistance myotrauma (disuse atrophy) represents the effect of excessive respiratory support and reduced respiratory drive/effort, resulting in disuse atrophy and weakness. It is the best documented form of myotrauma, in both experimental studies [30,31] and clinical studies reporting histological [24,32,33], functional [7], and imaging evidence of disuse atrophy [23,34,35,36]. It is important to emphasize that disuse atrophy can occur even in assisted ventilation modes [37].Under-assistance myotrauma (load-induced, concentric contraction) is a recognized form of muscle injury that occurs when ventilatory support is insufficient or unable to reduce the load on the diaphragm during increased respiratory drive (load-induced diaphragm injury). The persistence of high levels of respiratory efforts during under-assistance results in high muscle tension during concentric contraction [23,38]. In experimental models, the diaphragmatic injury is characterized by an inflammatory infiltration and the disruption of sarcomere and sarcolemma.Eccentric myotrauma (load-induced, eccentric contraction) occurs when there is contractile activity of the diaphragm during fiber lengthening instead of shortening. Eccentric contraction is observed in the expiratory phases of ventilation: (1) when the diaphragm exerts an expiratory brake to preserve the end-expiratory lung volume [39]; (2) in some forms of asynchrony (reverse triggering, early cycling, ineffective effort) [40,41,42].Expiratory myotrauma—in experimental models, it was observed that the use of high PEEP reduces the final expiratory length of the diaphragm and can cause loss of sarcomeres, resulting in longitudinal atrophy with impairment of the length-tension ratio of the diaphragm [43].

## 3. Diaphragm Dysfunction-Recognition

Considering the main mechanisms involved in myotrauma, more recently, the literature has been dedicated to the recognition and early monitoring of diaphragmatic dysfunction, allowing the appreciation and development of rational strategies with the potential to minimize or prevent VIDD [41] (Figure 1).

Any strategy aimed at protecting the diaphragm must examine its functional aspects, with a particular emphasis on monitoring the diaphragm’s effort [20]. The most grounded approach in recent literature is to maintain an adequate level of inspiratory diaphragmatic effort and curb different forms of AVP [20,41].

Various tools can be employed to monitor the function/activity of the diaphragm. However, the various peculiarities of these techniques make them more or less appropriate in critically ill patients [10,19,44]. The gold standard for assessing diaphragm function is based on measuring transdiaphragmatic pressure after magnetic stimulation of the phrenic nerves, which, however, is generally unavailable, thus requiring the development and implementation of more viable tests, among which diaphragmatic ultrasound emerged [19,20,41,42].

## 4. Diaphragm Ultrasound-DUS

Diaphragmatic ultrasonography has been shown to be a relevant tool for assessing diaphragm function that reached significance in critically ill patients due to its numerous advantages [45,46,47]: (a) The ultrasound devices needed to perform DUS are simple equipment, widely available in ICUs; (b) ultrasound machines are portable, which allow DUS to be performed at the bedside in a safe, repeatable and non-invasive way, without the need to transport the patient; (c) the improvement of the technique allows its fast, precise, reproducible, and repeatable execution.

The devices used to perform DUS have experienced advances (probes with better resolutions). In addition, the improvement and standardization of the technique allowed for the acquisition of reliable images of the diaphragm function [47,48,49].

### 4.1. Ultrasound of Diaphragm–Brief Review of the Technique

The diaphragm is a dome-shaped, musculo-fibrous membrane. It is the main inspiratory muscle. Evaluation of the diaphragm with ultrasound is based on its anatomy and function. During inspiration, when the diaphragm contracts, the muscle fibers shorten, thicken, and stiffen, displacing the entire musculotendinous structure of the diaphragm, lowering its dome, moving the abdominal contents caudally and expanding the lower thoracic cavity [47].

Using the ultrasound, the diaphragm may be explored through two acoustic windows [46,47,50] (Figure 2):-In the subcostal region, the diaphragm is seen as a deep curved structure that separates the thorax from the abdomen;-In the mid-axillary intercostal region, in the zone of apposition (ZA), the diaphragm is identified as a three-layer structure; a hypoechoic inner muscle layer surrounded by two hyperechoic outer membranes (the peritoneum and pleura).

Thus, during diaphragm inspiratory contraction, the ultrasound may observe: (a) the inspiratory caudal displacement (excursion) of the diaphragm in the subcostal region; (b) the end-expiratory thickness of the muscle (at rest) and the thickening and stiffening of the diaphragm during inspiration, in the ZA.

In critically ill patients, a unilateral assessment of the right diaphragm is acceptable unless unilateral dysfunction is suspected (requiring bilateral assessment) [48].

Recently, a meeting of experts sought to outline a consensus focusing mainly on various technical/methodological aspects of ultrasound of the diaphragm. This consensus is therefore an important reference to standardize and universalize the performance of diaphragm ultrasound and should be regularly consulted in the application of the method.

Excursion assessment: a low-frequency probe (3.5–5 MHz) was positioned below the right costal margin, between the midclavicular and anterior axillary line, directing medial-cranially and dorsally, tracking the posterior third of the right hemidiaphragm in a two-dimensional (2D) mode. Next, the M-mode exploration line was positioned as perpendicular as possible in relation to the diaphragm domus, in order to measure the movement of anatomical structures over time, and therefore, measure diaphragm mobility by placing calipers at the bottom and top of the diaphragmatic inspiratory slope [47,51,52].

During inspiration, the diaphragm descended in the craniocaudal direction, towards the probe [46,47,50]. Thus, during inspiration, absent or greatly reduced excursion (below lower thresholds) or movement contrary to the probe (paradoxical) indicated an abnormality (dysfunction).

In patients on ventilatory support, a brief disconnection of the ventilator (T-tube) or maintaining minimal levels of CPAP was necessary to assess diaphragmatic movement and thus eliminate passive inflation bias by ventilator pressure [46]. The downward displacement of the diaphragm may result from the force of the diaphragm contraction by itself, and also the passive lung inflation performed by the ventilator, indicating that diaphragm excursion should only be assessed during unassisted breaths [53,54].

Excursion can be measured during tidal breathing and during maximal inspiratory effort in cooperative patients (best excursion) in order to assess diaphragm dysfunction [55].

Diaphragm thickness assessment: a high frequency linear probe (>7–10 MHz) was positioned perpendicular to the lateral chest wall, between the midaxillary and anterior axillary line, between the 9th and 10th intercostal space (apposition zone–ZA). The diaphragm is seen usually at a depth of two to four centimeters from the skin, as a hypoechoic inner muscle layer surrounded by the peritoneum and pleura membranes. Diaphragmatic thickness was measured at the end of expiration (Tdi,exp) and inspiration (Tdi,insp) as the distance between the diaphragmatic pleura and the peritoneum using the 2D or M-mode [47,56,57]. Although around 1.5 mm has been reported as a lower limit for normal diaphragm thickness in healthy individuals [58], in very thin healthy women, thickness values of up to 1.2 mm can be observed [57]. Diaphragm thickening fraction (TF) is defined as the percentage change in diaphragm thickness during inspiration [(Tdi,insp) − (Tdi,exp)/(Tdi,exp)] × 100 [46,47,50].

Diaphragm thickness (Tdi) may depict atrophy, while the thickening fraction (TF) represents diaphragm inspiratory activity (effort) [59,60,61]. 

Reference values for diaphragmatic excursion, thickness and thickening in healthy individuals are described in the Appendix A.

### 4.2. Diaphragm Ultrasound in Respiratory Muscle Monitoring in Critically Ill Patients 

The following is an overview of the recent evidence on diaphragmatic and intercostal (IC) ultrasound assessment, focusing on clinical applications aimed at promoting diaphragmatic protective MV strategies (Table 1).

#### 4.2.1. To Diagnose Diaphragmatic Dysfunction

Diaphragm dysfunction can be identified with DUS. Diaphragm dysfunction is characterized on DUS as an abnormality on diaphragmatic excursion indicated by reduced mobility (less than the reference value), absent, or paradoxical (mainly seen in deep breathing and sniff maneuvers) [62]. In critically ill patients on MV, the most commonly used criterion to indicate diaphragmatic dysfunction is a diaphragmatic excursion < 10–11 mm [11,63] (Figure 3). 

Ultrasound-diagnosed DD is associated with adverse outcomes (longer MV and weaning times, as well as higher mortality). Abnormal diaphragmatic thickening fraction, defined as a diaphragmatic thickening fraction of <20%, has also been used to diagnose DD [64]. According to previous studies, the prevalence of DD has been found to be 28–38% among patients on MV eligible for an SBT [11,37], and 34% among patients on MV for prolonged periods [64]. Patients with DD seen on ultrasound had worse outcomes (prolonged MV, weaning failures and increased mortality) [11,63,64]. 

#### 4.2.2. Ultrasound of Diaphragm to Monitor Diaphragm Atrophy

Diaphragm atrophy is a component of VIDD, and has been shown in experimental [31,76,77] and clinical studies that demonstrated the presence of histological atrophy (defined as a decrease in diaphragm fiber cross-sectional area in diaphragm biopsy) even after few hours of MV [7,21,24,37].

Several studies have used the measurement of diaphragmatic thickness at the end of expiration (Tdi) as a non-invasive surrogate of diaphragmatic muscle mass, thus allowing the assessment of diaphragmatic atrophy in critically ill patients, especially those on MV [45,78]. In addition, repeated assessment of Tdi over the days of MV may indicate changes in diaphragm configuration, including muscle thinning, that may be indicative of atrophy [22,23,34,35,36].

The wide range of values to determine the lower limit of Tdi (between 1.6 and 2.3 mm) both in healthy individuals and in critically ill patients on MV in different studies [56,57,58,78,79] makes it difficult to establish a cut-off to define atrophy in a single measurement. Currently, there is more consistent evidence on the relevance of serial thickness monitoring to define the occurrence of atrophy, one of the markers of diaphragm dysfunction [23,34,35,36,80]. Using DUS, atrophy has been defined as a decrease of 10% or more of the diaphragm in serial measurements [23,80].

Goligher et al. [23] studied the evolution of diaphragm thickness over time during MV in 107 adult patients. They found that over the first week, mainly in the first 72 h of MV, diaphragm thickness decreased (more than 10%) in 44% of patients, and this atrophy was associated with low diaphragm contractile activity. Later on, these authors reassessed whether diaphragm atrophy developing during MV could impact clinical outcomes [22]. They found that atrophy was associated with prolonged ventilation and ICU admission, and a higher risk of complications.

Subsequently, other studies observed the relationship between atrophy and MV [35,36]. Schepens et al. [35] described that diaphragm atrophy (mean change in thickness of 32%) occurs quickly after the onset of MV in 54 ICU adult patients. The degree of atrophy was associated with the length of MV [35]. Zambon et al. [36] used ultrasound to measure the diaphragm thickness daily, from the first day of MV till discharge in 40 adult patients, in order to quantify diaphragm atrophy and identify risk factors. They observed a gradual reduction in diaphragm thickness (estimated as a daily atrophy rate) which presented a linear relationship with ventilator support ranging from −7.5% under controlled MV to +2.3% during spontaneous breathing.

Vivier et al. [80] studied a sample of critically ill patients to assess serial diaphragm thickness during the first five days of critical illness. They found that diaphragm atrophy (Tdi decrease of ≥ 10%) occurred in 48% of patients, and atrophy was associated with septic shock, organ failure, or invasive MV.

Regarding a single measurement of Tdi to assess atrophy, Dot et al. [66] investigated whether expiratory thickness (Tdi) measured by ultrasound was able to predict diaphragm atrophy, defined by a decrease in diaphragm fiber cross-section area (CSA) obtained through diaphragm biopsy in 35 ventilated organ donors. They found that all of the donor group presented atrophy (reduced CSA), but decreased Tdi was observed in 74%. Diaphragm Tdi measurement demonstrated a high positive predictive value (96%) but a low negative predictive value (17%) for determining the presence of diaphragm atrophy, indicating that the diaphragm Tdi was able to detect atrophy, but cannot rule it out completely.

However, even a single Tdi measurement at the early stages of critical illness can provide relevant information. Recently, Sklar et al. [78] demonstrated that in critically ill patients on MV, a low diaphragmatic muscle mass (baseline Tdi of 2.3 mm or less) measured by ultrasound in the first 36 h after intubation, was associated with adverse clinical outcomes.

#### 4.2.3. DUS to Monitor Effort–Diaphragm Protective Ventilation

Our understanding of the mechanisms of ventilator-induced diaphragmatic dysfunction (VIDD) has increased in recent years. Research now focuses on methods to monitor diaphragm function in parallel with efforts to prevent VIDD. The institution of assisted ventilation (“spontaneous breathing”) as early as possible seems to meet the efforts to prevent VIDD, but poses challenges to also comply with MV with “lung protection”. 

Also, as already demonstrated in previous studies, diaphragmatic atrophy due to disuse can occur even in assisted ventilation [37,67,77] indicating that simply triggering the ventilator is not enough to prevent diaphragm atrophy. In addition, clinical variables and ventilator settings are not able to detect contractility and risk of diaphragm atrophy. [67].

At this moment, the titration of the inspiratory effort and the synchronization in the patient-ventilator interaction seem to provide the best evidence in the simultaneous protection of the lung and the diaphragm, although the efficacy of such strategies is mainly supported by physiological and epidemiological data [19,20,41,42,44].

Recent studies are now in course to assess the feasibility of strategies on titrating ventilation and sedation, in order to optimize the respiratory effort to achieve lung and diaphragm protective ventilation [81]. In a recent trial [82], inspiratory support was titrated based on a predefined “diaphragm-protective” range of transdiaphragmatic pressure swings (3–12 cm H_2_O). This strategy, based on patient breathing effort, increased the time that patients had this predefined “diaphragm-protective” effort without compromising tidal volumes and transpulmonary pressures. 

Although in these previous studies the respiratory effort titration was based on the monitoring of respiratory pressures, some authors proposed that the DUS can be useful in assessing the contractile activity of the diaphragm, and therefore also in assessing the respiratory effort [19,20,22,44].

Assessment of the diaphragm thickening fraction (TF) as a surrogate of contractile activity of the diaphragm (the level of respiratory effort) is a reasonable approach [22,49]. Adjustment of ventilatory support to avoid excessive or insufficient ventilatory assistance may mitigate MV damage on the diaphragm. The titration of ventilatory support to maintain the contractile activity of the diaphragm (effort) within physiological limits may prevent changes in diaphragm configuration [22,23] (Figure 4).

Although the optimal level of diaphragmatic activity is uncertain, Goligher et al. [23] showed that diaphragm thickness was stable over time, at levels of contractile activity that are typically observed in healthy subjects during resting tidal breathing. In this study, diaphragm contractile activity directly influenced the change in diaphragm thickness over time. For the first time, it was demonstrated that diaphragm thickness (Tdi) decreased over time during lower contractile activity (TF) while Tdi increased during higher contractile activity (TF). In a subsequent study of this group, a TF between 15–30% during the first few days of MV was associated with stable muscle thickness and a shorter duration of ventilation [22].

These findings along with studies that observed the existence of a correlation between TF and diaphragm effort [65,68] serve as an argument for assuming that maintaining a TF of 15–30% may indicate a safety threshold for titration of effort during MV [49].

However, it should be emphasized that the use of ultrasonography of the diaphragm, particularly the inspiratory thickening fraction (TF), as a surrogate for diaphragmatic effort, is still debated in the literature.

Transdiaphragmatic pressure (Pdi) represents the specific muscle strength of the diaphragm, and its measurement has been considered the gold standard for diaphragm function, particularly after the bilateral magnetic stimulation of phrenic nerves [83]. However, Pdi measurement has been neglected due to its problematic use in clinical settings (unavailability and invasiveness of esophageal and gastric catheters, expertise in interpretation) [84]. To overcome this, ultrasound has been proposed as a non-invasive tool to assess diaphragm function [45]. During inspiration, diaphragmatic contraction promotes the shortening of muscle fibers that can be observed in the apposition zone by ultrasound and quantified. The diaphragm thickening fraction (TF) is the magnitude of the increase in diaphragm thickness during inspiration [46]. Inspiratory thickening of the diaphragm has been proposed as a reflection of the magnitude of diaphragm effort [54], which can be corroborated by studies that evaluated diaphragm thickness in individuals with diaphragmatic paralysis, in which no thickening was seen on ultrasound in subjects who did not recover from diaphragmatic paralysis [85].

However, there are controversial findings regarding the ability of TF to estimate diaphragmatic effort. Vivier et al. [68] investigated whether the thickening fraction (TF) could be indicative of respiratory muscle effort in 12 patients who required planned non-invasive ventilation (NIV) after extubation. The work of breathing in this study was indicated by the transdiaphragmatic pressure-time product (PTPdi). Patients were evaluated during spontaneous breathing and during NIV at three levels of pressure support (5, 10 and 15 cm H_2_O). The authors found a significant correlation between TF and PTPdi, indicating that changes in diaphragm thickening correlate with the diaphragmatic work of breathing.

Umbrello et al. [54] investigated the performance of diaphragmatic contractile activity seen on ultrasound (diaphragm excursion and thickening fraction, TF) in comparison with classic indices of inspiratory muscle effort (esophageal pressure-time product (PTPes) and diaphragmatic pressure-time product (PTPdi)), during assisted MV in 25 patients admitted to the ICU after major elective surgery who fulfilled criteria for a trial of spontaneous breathing with pressure support (PS) ventilation at different levels of PS (0, 5 and 15 cmH_2_O). The authors observed that with increasing PS levels, parallel reductions were found between diaphragm TF, PTPes and PTPdi during tidal breathing. Diaphragm TF significantly correlated with PTPes and PTPdi. There was no correlation of any index with the excursion of the diaphragm.

Goligher et al. [45] investigated in five healthy subjects (validation study) the association between the diaphragm thickening fraction with the electrical activity of the diaphragm (EAdi). The TFdi and EAdi were positively correlated. At inspiratory volumes below 50% of inspiratory capacity, passive insufflation did not cause diaphragm thickening, indicating that at clinically relevant inspiratory volumes, diaphragm thickening reflects muscle contraction and not passive insufflation.

In order to estimate reliable measures for assessing respiratory effort, Umbrello et al. [73] investigated variations in esophageal (Delta Pes) and transdiaphragmatic (Delta Pdi) measurements and its derivations (esophageal pressure-time products (PTPes), transdiaphragmatic (PTPdi) and work of breathing (WOB)) in comparison with the variations in diaphragm thickening (TF) and parasternal intercostal thickening (TFic) in 21 critically ill patients on MV with pressure support (PS). Measurements were performed at three PS levels: baseline, 25% and 50% reduction. The authors found that TF was only weakly correlated with esophageal PTP, but this correlation improved after excluding patients with diaphragm dysfunction (as defined by the Gilbert index).

Unlike these previous studies [45,54,68,73], the relationship between thickening fraction and respiratory effort was not observed in other papers, as presented below. 

Oppersma et al. [86] investigated the validation of speckle tracking (strain and strain rate) to assess the inspiratory diaphragmatic effort compared to the diaphragm thickening fraction, in 13 healthy volunteers during an inspiratory overload protocol (randomized from 0 to 50% of maximal inspiratory pressure). Diaphragm electrical activity (EAdi) and transdiaphragmatic pressures (Pdi) were also recorded. Strain and strain rate increased with progressive diaphragm loading. Both strain and strain rate were highly correlated with Pdi and EAdi, but TF was not influenced by inspiratory load variations and did not correlate with EAdi and/or Pdi. 

Poulard et al. [87] investigated the global and intraindividual relationship between the diaphragmatic thickening fraction (TF) and transdiaphragmatic pressure (Pdi; the PTPdi was calculated) in 14 healthy subjects (in whom they applied variations in the external inspiratory threshold load) and 25 mechanically ventilated (MV) patients at various pressure support settings. They found that in healthy subjects, both changes in Pdi and PTPdi were moderately correlated with TF. In MV patients, changes in Pdi and TF were weakly correlated, indicating that TF was not able to infer the pressure output.

Recently, Steinberger et al. [88] investigated whether the diaphragmatic excursion (DE) and the thickening fraction (TF) measured by ultrasound could reflect the variation in esophageal pressure (Delta Pes) during continuous positive airway pressure (CPAP) ventilation in 46 patients hospitalized for COVID-19 pneumonia. The authors did not find any association between Delta Pes and DE or TF, although only a moderate correlation was observed between variations in Pes oscillation and variations in TF between PEEP and ZEEP.

These controversial findings may arise from many points. (1) Heterogeneity of the population studied: Goligher and Oppersma [45,86] investigated healthy individuals, while Poullard [87] assessed a mixed sample of healthy and mechanically ventilated patients; however, the other studies assessed specific populations of patients (patients requiring planned NIV after extubation [68], patients admitted to the ICU after major elective surgery who met criteria for an SBT [73] and patients with COVID-19 pneumonia on CPAP [88]). (2) The studies have different conditions to test effort: studies applied different protocols of inspiratory loading in healthy individuals [86,87] who could adopt diverse patterns of respiratory muscle recruitment. (3) Regarding the studies on patients, they could be on various configurations of lung volume and chest wall compliance [54,68,73]. (4) In critically ill patients, diaphragm dysfunction (DD) may influence the relationship between diaphragm thickening and inspiratory effort [73], but DD was not tested on the study of Steinberg [88]. (5) It is largely recognized that the thickening fraction associates with a wide range of PTPdi values, which may signify that the degree of diaphragmatic contraction and corresponding work of breathing may vary between subjects [86].

Although these controversies indicate that caution should be taken in interpreting TF as a substitute for respiratory effort, we believe that Goligher’s findings that correlate changes in diaphragm configuration (estimated by Tdi) with diaphragmatic contractile activity (estimated by thickening fraction) may indicate that TF offers a plausible alternative for estimating respiratory effort, whose monitoring is essential for the prevention of diaphragmatic myotrauma [22,23].

## 5. Patient–Ventilator Asynchrony (PVA)

Patient–ventilator asynchrony (PVA) can be defined as a mismatch between patient effort/demand and ventilator-delivered breaths (regarding time, flow, volume, or pressure) [89].

PVA is not uncommon and may occur in up to a quarter of mechanically ventilated patients [49,90,91], but is often ignored and underestimated.

PVA is associated with adverse effects such as dyspnea, increased work of breathing, auto-PEEP, worsening of pulmonary gas exchange, decreased quantity, and quality of sleep, leading to worse outcomes (increased use of sedation and neuromuscular blockade, prolonged MV, increased duration of MV, and increased mortality [90,92,93,94,95]. 

PVAs most commonly result from a mismatch in the patient-ventilator interaction during the trigger of the inspiratory cycle phases (ineffective effort, self-trigger, and double trigger) and the end of the inspiratory phase (premature or delayed cycling), in addition to an imbalance in the inspiratory flow (insufficient or excessive) [89].

PVA potentially impacts on diaphragm function and has been proposed as one of the mechanisms linking MV to diaphragm weakness, i.e., myotrauma [19]. 

Optimal patient-ventilator interaction, without asynchrony, would theoretically be ideal for the diaphragm [89]. More recently, it has been established that performing MV, with lung and diaphragm protection, requires the monitoring of respiratory effort [41]. The DUS can thus be a valuable tool for providing the assessment of the respiratory effort (by measuring the contractile activity of the diaphragm represented by the thickness fraction), in addition to providing the identification of structural alterations in the muscle [4].

As previously described, the main mechanisms associated with myotrauma are: over-assistance leading to respiratory effort suppression and disuse atrophy, under-assistance associated with high effort and load-injury during the inspiratory concentric contraction, eccentric contraction during the expiratory phase, and the longitudinal atrophy. 

PVA-associated muscle injury is mainly related to eccentric contraction [4]. In certain types of asynchrony, the patient’s inspiratory efforts (such as ineffective effort) may occur during the expiratory phase, determining eccentric or plyometric contractions (contractions that occur during the fiber stretching) and causing injury to the diaphragm muscle fibers [96,97].

Although the visual inspection of the airflow, volume, and pressure signals on the mechanical ventilator display may point to asynchrony, PVAs may go unnoticed in this method [1,12].

The gold standard for detecting PVA is esophageal pressure monitoring (Pes), providing an accurate inspection of the patient’s inspiratory muscle effort and corresponding ventilatory supply. However, monitoring of Pes remains limited because it is commonly unavailable and has technical pitfalls (e.g., tight control of esophageal balloon volume and location) [89,98].

Diaphragmatic electrical activity (EAdi) is another tool indicated for monitoring PVA, and is also considered a reference technique [99], but with similar limitations to Pes to use in daily practice (unavailability, invasiveness, specialized training for interpretation) [49]. EAdi monitoring can detect trigger delays, early and late cycling, auto-triggering, double-triggering, and wasted efforts.

Considering that the inspiratory diaphragmatic activity (contraction) determines the changes in pleural pressure, it has been argued that DUS may be used to simultaneously monitor the function of the diaphragm and the breathing pattern in order to detect PVA [48,49].

Soilemezi et al. [100] employed simultaneous recordings of diaphragmatic displacement (DUS) and airway pressure to assess patient–ventilator asynchrony in three patients. The authors describe in this method three representative cases of asynchrony detection, demonstrating: (a) a patient with double triggering (the ventilator delivers two consecutive breaths for a single patient effort); (b) a patient with ineffective efforts (due to excessive ventilatory assistance and long insufflation times with shortening of the patient’s expiratory phase, which prevents the patient’s next inspiratory effort from exceeding intrinsic PEEP and activating the ventilator); (c) a reverse-triggered patient (when inspiratory efforts are triggered directly by mechanical breaths). The major limitation of this technique is the unavailability of continuous recordings. 

Simultaneous observation of TF and Paw may indicate an auto-trigger PVA, when an assisted breathing cycle is observed without detection of inspiratory thickening [48,49].

Vivier et al. investigated the performance and accuracy of a new method based on the ultrasound analysis of diaphragm excursion (DE) or thickening combined with airway pressure to detect PVA. They found that this method (especially diaphragm thickening) was more accurate than flow and pressure waveform analysis alone, and had better feasibility than diaphragm EMG for detecting patient-ventilator asynchronies [101].

It should be noted that although DUS might be a reasonable alternative to detect most types of PVA [49], further studies are needed to determine its exact role [46,100].

## 6. Role of Ultrasound of the Diaphragm in the Assessment of Weaning

Weaning from MV is a process of gradual reduction of ventilatory support, configuring a challenge to the respiratory muscles, notably the diaphragm. Unsuccessful weaning from MV have worse outcomes increasing morbidity and mortality [102,103].

The main cause of weaning failure is an imbalance between the load and capacity of the respiratory system [104]. Identifying patients who are suitable or not for weaning avoids, on the one hand, the unnecessary use of MV with its inherent risks, and on the other hand, prevents premature weaning and its complications. For this characterization, clinical criteria (“readiness testing”) are employed in order to identify patients who are ready or not to wean from MV [103]. However, readiness testing can sometimes point to uncertainty regarding the patient’s ability to wean and, in these circumstances, the use of physiological tests and predictors of weaning can help identify patients who are ready or not to wean [104].

Diaphragm function assessed on ultrasound has been used as a predictor of weaning, primarily as a diagnostic test to predict extubation success [50,105].

Numerous studies have used the DUS to assess the prediction of weaning, and there are even meta-analyses and systematic reviews on the subject in the literature [50,105,106,107].

The earliest studies suggested that a successful extubation or weaning failure was likely with diaphragm excursion cut-off values of 11–14 mm [11] and when the diaphragm thickening fraction was lower than 30–36% [108].

However, recently Vivier et al. [80] assessed the accuracy of diaphragmatic excursion and thickening to predict extubation. DUS was performed before extubation, in a T-tube, in 191 patients who successfully underwent an SBT. Diaphragmatic dysfunction, defined as excursion < 10 mm or thickening < 30%, was not associated with an increased risk of extubation failure. Vetrugno et al. [109] also demonstrated recently that diaphragm thickening fraction (TF) was not predictive of weaning failure from MV, in 57 COVID-19 patients undergoing weaning from MV. 

Regarding the value of DUS to predict weaning from MV, it must be assumed that the results are controversial, and general conclusions are not allowed at this moment.

This controversy seems to be mainly related to the heterogeneity regarding the population and study design, and on the characterization of the variables under test. There are marked variations in:

(1) the timing of the ultrasound assessment in relation to the spontaneous breathing test (SBT), patient position during the SBT,

(2) the standardization of the diaphragm evaluation technique with ultrasound in patients on ventilatory support (PEEP levels, pressure support),

(3) varying definitions of SBT failure, weaning failure, or extubation failure, which were not standardized across studies, making comparison between outcome measures difficult.

Furthermore, a substantial proportion of patients with diaphragmatic dysfunction seen on DUS could be successfully weaned from the ventilator [10] suggesting that in patients undergoing a successful SBT, extubation appears to be determined by factors other than diaphragm function [49].

Although diaphragmatic ultrasound is a promising diagnostic tool, to determine its role in predicting weaning, additional studies are needed to standardize [49] some relevant points in the technical aspect [48] and also regarded to the clinical question [110], suggesting that:-To identify patients at high risk of weaning failure, DUS is performed before SBT.-To predict the weaning outcome or to diagnose the cause of weaning failure, DUS is best performed after initiation and/or completion of an SBT.

## 7. Novel Techniques and Future Developments for Functional Imaging and Quantification of Tissue Properties 

### 7.1. Tissue Doppler Imaging 

Tissue Doppler Imaging (TDI) is an ultrasound technique that detects the change in the frequency of ultrasound signals reflected by moving structures, already widely used to assess cardiac function but only more recently applied to assess the diaphragm in adults [111] and neonates [112].

Although scarcely studied, recently Soilemezi et al. [113] presented interesting results, studying healthy volunteers (*n* = 20) with the aim of describing the pattern, reporting normal values, and assessing the reproducibility of diaphragmatic TDI waveform velocities during contraction (including peak contraction velocities (PCV) and relaxation (including maximal relaxation rates (TDI-MRR)). Also in this study, the authors recorded the pattern and values of diaphragmatic TDI in ICU intubated patients (*n* = 106) elected for a spontaneous breathing test, comparing cases of successful and unsuccessful weaning, and, in a subgroup of these patients (*n* = 24), they correlated TDI measurements with parameters derived from transdiaphragmatic pressure (Pdi). The results indicated the patterns of contraction and relaxation of the TDI diaphragm in normal subjects and in patients with successful and unsuccessful weaning. It was a notable finding that weaning failure patients exhibited significantly higher PCV and TDI-MRR velocities compared to healthy participants and weaning success patients, such that the higher the values of PCV and TDI-MRR during weaning, the greater the likelihood of weaning failure. In addition, in the subgroup of patients evaluated with Pdi, high correlation coefficients of TDI PCV with peak Pdi and PTPdi, and Pdi-MRR and TDI-MRR were demonstrated. These findings offer insights into the physiological behavior of patients with weaning failure using a non-invasive technique.

Cammarotta et al. [111] investigated whether the diaphragmatic function assessed by TDI (to measure the excursion velocity and acceleration) could help to discriminate the success of extubation (after 48 h) among 100 adult subjects who passed a 30-min spontaneous breathing trial (SBT). The TDI measurement was performed at the end of the SBT. They found that: (1) TDI parameters (collected at the end of an SBT) discriminated against individuals likely to exhibit extubation failure within 48 h after extubation (in these individuals the dTDI variables were significantly higher when compared to those who were successfully extubated). Inspiratory peak and mean velocity were good predictors of extubation failure after a successful SBT; TDI assessment was feasible and reproducible. 

Although the preliminary results are very significant, it is important to point out that much progress is still needed in the standardization of the method that includes the determination of the diaphragmatic movement point of interest (ROI); influence of other factors, such as compliance of the rib cage and activation of accessory inspiratory and expiratory muscles [114].

### 7.2. Speckle Tracking Ultrasound (STUS) 

Traditional methods to assess diaphragmatic function on ultrasound, such as measuring diaphragmatic excursion, diaphragm thickness and thickening fraction (TF), although consolidated, are limited to measuring the deformation of diaphragmatic muscle tissue in the transverse axis and therefore do not assess the ‘longitudinal’ muscle shortening, that is, the movement in the direction of the contracting muscle fibers.

Speckle tracking ultrasound (STUS) is a new method of analysis, already widely used in echocardiography [115], which allows observing the contraction of muscle fibers in the direction of movement (longitudinal). 

A speckle represents a region of muscle tissue whose gray pattern on ultrasound remains relatively constant. The analysis of the STUS software is based on following (to track) a group of speckles during the contractile cycle, measuring their displacement and deformation (movement of the speckles in relation to each other). The degree of deformation is known as ‘strain’ while “strain rate” measures the deformation velocity. Because STUS tracks specific regions of interest it is considered relatively angle independent and tracking can help differentiate passive movement from active contraction.

The applicability of using STUS to investigate diaphragm movement was tested in different scenarios. 

Hatam et al. [116] analyzed diaphragm deformation (transverse strain) in STUS as a parameter for respiratory load in healthy individuals (*n* = 13) submitted to NIV in CPAP (CPAP 5, PSV 5, 10 and 15) or unsupported (spontaneous breathing) and compared with inspiratory thickening fraction (TF). They found that both TF and transverse strain significantly increased under CPAP and PSV and that transverse strain correlated well with TF. 

Oppersma et al. [86] evaluated the validity of speckle tracking ultrasound (strain and strain rate%) to quantify diaphragm contractility in healthy volunteers (*n* = 13) submitted to an inspiratory overload protocol (randomized from 0 to 50% of the maximum inspiratory pressure). Diaphragm electrical activity (EAdi), transdiaphragmatic pressures (Pdi) and thickening fraction were also recorded. They found that strain and strain rate, but not TF, increased with progressive loading of the diaphragm. Both strain and strain rate were highly correlated with Pdi and EAdi, but not TF. In this study, STUS was superior to conventional ultrasound techniques (TF) for estimating diaphragmatic contractility under inspiratory threshold loading.

Orde et al. [117] evaluated the applicability of STUS parameters (longitudinal strain) to analyze diaphragmatic contraction in healthy adults (*n* = 50) in comparison with the evaluation of diaphragm thickness, thickening fraction (TF) and excursion (diaphragm caudal displacement). They found a moderate correlation between right diaphragmatic longitudinal strain and TF, and a weak correlation between strain and excursion. Inter- and intra-rater variability was satisfactory for TF, excursion and strain.

Xu Q et al. [118] evaluated the feasibility and reproducibility of quantification of diaphragm longitudinal strain (DLS) by STUS (study A: in 25 healthy subjects and 20 mechanically ventilated patients), and aimed to determine whether the maximum DLS can be used to predict outcomes weaning (study B: multicenter retrospective, 96 patients elected for weaning, of which 56 were successfully weaned). Intra- and inter-operator reliability was good to excellent under conditions of eupnea, deep breathing and MV. DLS exhibited a fair linear relationship with diaphragmatic thickening fraction (TF) and diaphragmatic excursion (DE). For the prediction of successful weaning, the AUC of ROC curves of DLS was very similar to TF, but not superior to the rapid shallow breathing index. 

Fritsch et al. [119] evaluated the feasibility of STUS for an evaluation of the diaphragm in a group of 20 patients after a coronary artery bypass graft procedure, in order to detect diaphragm dysfunction that would manifest as a decrease in the strain and strain rate parameters. Each patient received three ultrasonographic assessments: (1) preoperatively, (2) within 24 h, and (3) within 48 h after extubation. They found that the deformation of the diaphragm (strain) was reduced for about 24 h, but the contractile velocity (strain rate) remained stable or even increased. This increase in the strain rate was attributed to a change in the recruitment pattern of diaphragmatic fibers, which may reflect an increasing activation of type II fibers.

Although there are promising results, STUS lacks validation in a large study. In addition, the software for STUS analysis was designed to determine myocardial deformation, and they are “closed boxes” that need to be broken. Many studies performed offline analyses, demanding considerable time, and making applicability in a clinical situation difficult. Additional studies involving STUS in subjects with abnormal diaphragmatic function and/or critical illness, and comparison with electromyography or transdiaphragmatic pressure measurements may be needed.

### 7.3. Shear Wave Elastography 

Shear wave elastography (SWE) is a new technique recently applied to assess the diaphragm, which consists of creating a source of vibration (focused acoustic impulse beam generated by the ultrasound probe), resulting in tissue deformation and generating a shear wave that can be quantified (converted into Shear Modulus, SM).

The SM expresses tissue biomechanical information and, therefore, is innovative for translating qualitative information about the properties of the diaphragm muscle. A higher SM indicates greater tissue stiffness.

In two previous studies of healthy subjects, SWE reproduced the diaphragmatic contractile activity.

Chino et al. [120] studied diaphragm function with SWE (shear modulus, SM) in 14 healthy male subjects at various levels of submaximal inspiratory tasks (at 15, 30, 45, 60, and 75% of the maximal inspiratory mouth pressure). They found that the diaphragm SM increased along with rising mouth pressure, but interestingly, the rate of SM increase decreases. Similar findings were observed by Bachasson et al. [121] who evaluated whether SWE (diaphragm shear modulus quantification, SMdi) could be used to assess diaphragm function compared to the measurement of transdiaphragmatic pressure (Pdi) in 15 healthy volunteers undergoing an inspiratory load protocol. They found that the mean Pdi was correlated with the mean SMdi and that changes in diaphragm stiffness assessed by SWE reflected changes in Pdi.

Aarab et al. [122] evaluated changes in the shear modulus (SM) and the relationship with changes in diaphragm thickness in patients during their ICU stay and also, in an experimental model, they characterized histological and force-producing changes in the diaphragm. As a translational study, it comprised a sample of critically ill patients (*n* = 102, of which 88 were mechanically ventilated) and mechanically ventilated piglets (in which transdiaphragmatic pressure and diaphragmatic biopsies were collected). The study observed a heterogeneous pattern of diaphragm configuration changes. A decrease or increase of more than 10% from the baseline was common in patients for thickness (86%) and SM (92%). Specifically, the SM remained unchanged in 8%, increased in 51%, and decreased in 41% of patients. Changes in the SM were inversely correlated with changes in diaphragm thickness in critically ill patients. A notable finding was that in the experimental model, the reduction in the SM was associated with diaphragm atrophy, lipid accumulation, and reduced strength. 

Although Aarab’s [122] findings are notable and seem to corroborate the concepts of diaphragmatic myotrauma, it is still necessary to extend SWE studies by standardizing the technique (ideal probe location, influence of lung volume, moment of measurement of MS in relation to the respiratory cycle) and expanding studies in different models and scenarios for a pathophysiological understanding of the method [123].

## 8. Limitations

Although ultrasonography has advantages when compared to other methods of assessing diaphragm function, its limitations must be considered for proper interpretation. Patient positioning [56] and the operator’s expertise in performing the ultrasound are important in acquiring and interpreting the images [124]. However, studies that have tested the increase in the appropriate skills to perform ultrasound through training have described a learning curve considered fast, with satisfactory results [108,125], especially when they involve a theoretical-practical combination, as recently observed by Garofalo et al. [126] and including critical care patients [127]. Regarding the concerns about patient positioning, although in healthy patients varied postures can interfere with the thickness of the diaphragm [128], Baldwin et al. [79] observed good reproducibility (intraclass correlation coefficient (ICC) = 0.990, 95% confidence interval: 0.918–0.998) in measuring the end-expiratory diaphragm thickness in a semi-recumbent position, as usual in critically ill patients. Considering the concern about variations in diaphragm thickness along the surface of the muscle [129,130] in serial measurements, demarcating the placement of the ultrasound probe can improve measurement consistency (ensure repeatability) [23]. It is recognized that there is greater difficulty in visualizing the left diaphragm due to the smaller window of the spleen [23] and ICU patients [45]. However, it should be noted that many critical patients have pleural (effusions) and/or pulmonary (consolidations or atelectasis) alterations that, contrary to what one might imagine, facilitate the identification of hemidiaphragms [46]. In critically ill patients, a recent expert consensus indicates that unilateral assessment of the right hemidiaphragm is an acceptable substitute for the entire diaphragm, except in cases of suspected unilateral involvement, when then there is a need for bilateral assessment of the diaphragm [48]. Ultrasound can also be limited due to a poor acoustic window for the visualization of the diaphragm in obese patients [47,131,132]. Finally, although a correlation between the thickening fraction and inspiratory strength has been observed in healthy subjects [57], the DUS does not provide a direct measure of ventilatory force, probably because the other inspiratory muscles of the ribcage may partially contribute to ventilation, interfering with the contribution of the diaphragm [51]. Even so, the assessment of diaphragmatic function with ultrasound has demonstrated numerous important clinical implications related to the outcomes of critically ill patients. As the tool seems to be in constant evolution, a meeting of specialists recently registered the main observations about the ultrasound technique that must be appreciated for the correct interpretation, aiming to overcome the main limitations [48]. Not all the cited works are homogeneous in terms of their study design, standardization of the method, in the selection of patients, or compared with a reference gold standard method.

## 9. Future Directions

Although ultrasonography has evolved significantly in recent times, there is still a long way to go for its wide use in clinical practice. Regarding critical illness, it is expected that potential improvements in transducers and in machine presets can improve the visualization and measurement of variables. In addition, it is essential that we have more studies that contrast sonographic findings of the diaphragm with:-Histological findings of the muscle,-Interaction with other respiratory muscles,-Reference tools for PVA detection, and-Baseline measures of respiratory effort (such as work of breathing, PTP) and fatigue (such as TTI and relaxation rate).

Also, a technical improvement that allows automated image acquisition can make it a genuine monitoring tool.

## 10. Conclusions

Diaphragm dysfunction is frequent and is related to many adverse outcomes in critically ill patients on MV. Diaphragm protection is a recent concept based on strategies aimed at preventing myotrauma. Monitoring diaphragmatic function and identifying diaphragmatic dysfunction are fundamentals of this protection strategy. Diaphragm ultrasonography may represent a possible complementary, promising tool which could be useful for monitoring diaphragm function at the bedside, potentially allowing you to adapt ventilator settings to provide protective diaphragm ventilation. Homogeneous prospective studies are needed in terms of study design, standardization of the method, in the number and selection of patients, and which all are compared with a reference gold standard method.

## Figures and Tables

**Figure 1 diagnostics-13-01116-f001:**
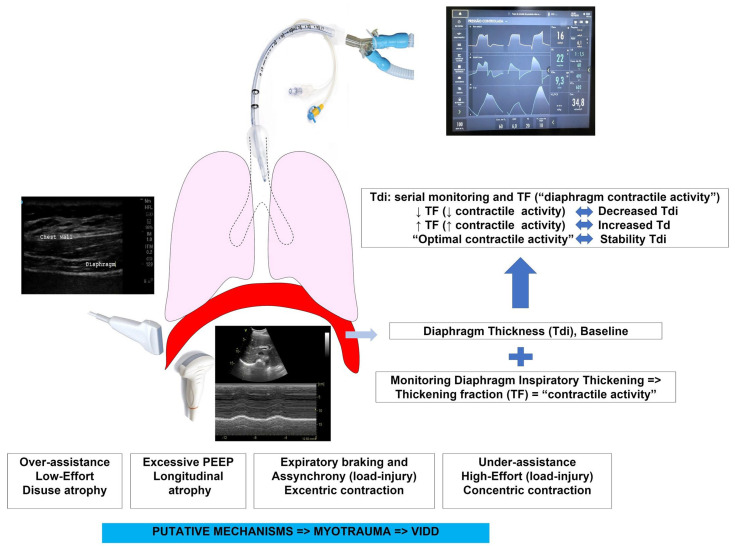
Possible rationale use of ultrasound to monitor diaphragm function in critically ill patients on mechanical ventilation (monitoring effort, assessing atrophy, assessing PVA). TF—thickening fraction; Tdi—diaphragm thickness; PEEP—positive end-expiratory pressure; VIDD—ventilator-induced diaphragmatic dysfunction.

**Figure 2 diagnostics-13-01116-f002:**
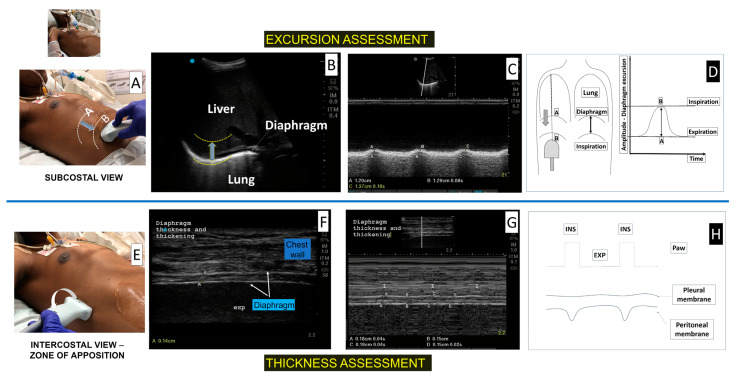
Diaphragm ultrasound to assess excursion and thickness – technique description in a critical care patient on mechanical ventilation. (Panel **A**): Probe positioning on subcostal area. (Panel **B**): Bidimensional view displaying the diaphragm as a white hyperechoic curved line (yellow traced) separating lung from liver. (Panel **C**): M-Mode view displaying a normal excursion during tidal breathing after a brief disconnection of the ventilatory support. (Panel **D**): Graphic illustrative of the excursion measurement. (Panel **E**): Probe positioning on the intercostal area (zone of apposition, ZA)—after positioning the linear probe perpendicularly to the costal wall, we rotated the probe (counterclockwise) to obtain the corresponding ultrasound image on the right. (Panel **F**): Bidimensional view displaying the diaphragm as a hypoechoic structure surrounded by 2 hyperechoic lines (pleural and peritoneal membranes). (Panel **G**): M-Mode view displaying a normal inspiratory thickening of diaphragm during tidal breathing in a critical care patient on mechanical ventilation (I = inspiration; points A and C represent end-inspiratory diaphragm thickness; points B and D represent end-expiratory diaphragm thickness. (Panel **H**): Graphic illustrative of the thickness and thickening measurement.

**Figure 3 diagnostics-13-01116-f003:**
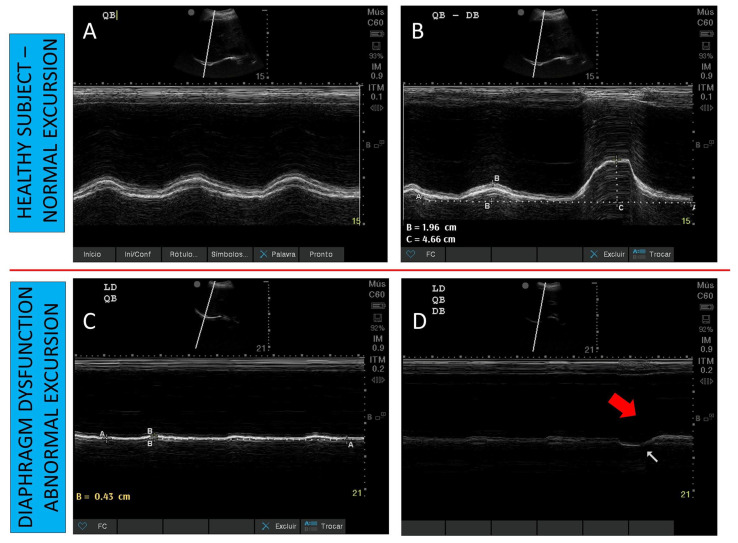
Diaphragm ultrasound to monitor dysfunction according to diaphragmatic excursion. (Panel **A**): Normal excursion during tidal breathing in a healthy subject. (Panel **B**): Normal excursion during tidal and deep breathing in a healthy subject. (Panel **C**): Reduced excursion during tidal breathing in a patient with diaphragm dysfunction. (Panel **D**): Absent excursion during tidal breathing, and paradoxical excursion during deep breathing (red arrow) in a patient with diaphragm dysfunction.

**Figure 4 diagnostics-13-01116-f004:**
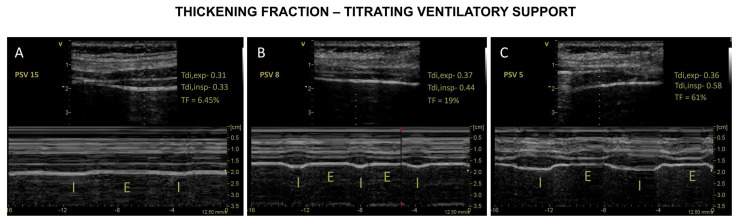
Diaphragm ultrasound to monitor effort. Titrating effort according to thickening fraction (TF). Panel **A**—Reduced TF suggestive of over-assistance (PSV 15 cm H_2_O). Panel **B**—“Ideal” TF suggestive of appropriated effort (PSV 8 cm H_2_O). Panel **C**—Increased TF suggestive of under-assistance (PSV 5 cm H_2_O).

**Table 1 diagnostics-13-01116-t001:** An overview of recent evidence on diaphragmatic and intercostal ultrasound assessment, focusing on clinical applications aimed at promoting diaphragmatic protective mechanical ventilation strategies. ICU—Intensive Care Unit; DE—diaphragmatic excursion; Pdi—transdiaphragmatic pressure; MV—mechanical ventilation; DD—Diaphragm Dysfunction; SBT—Spontaneous breathing trial; PSV—pressure support ventilation; OR—odds ratio; ARF—acute respiratory failure; TF—thickening fraction; ICUAW—intensive care unit acquired weakness; Ptr,stim—twitch tracheal pressure; ACV—Assist control ventilation; US—ultrasound; MRC—Medical Research council; Tdi—Diaphragm thickness; OTI—orotracheal intubation; Pdrive—driving pressure; CMV—controlled mechanical ventilation; ΔTdi/d—Daily atrophy rate; SD—standard deviation; CPAP—Continuous Positive Airway Pressure; CSA—cross-sectional area; PPV—positive predictive value; DUS—diaphragmatic ultrasound; RR—respiratory rate; Pes—esophageal pressure; Pga—gastric pressure; NIV—noninvasive ventilation; SB—spontaneous breath; PTPdi—diaphragm pressure—time product; EAdi—diaphragmatic electrical activity; AMV—assisted mechanical ventilation; PTPes—esophageal pressure–time product; EXdi—diaphragmatic excursion; C—intercostal; RV—residual volume; FRC—functional residual capacity; TLC—total lung capacity; IC—intercostal; COPD—chronic obstructive pulmonary disease; FEV1—Forced Expiratory Volume; CT—Computerized Tomographic; WOB—work of breathing.

Author, Year	Methods	n	Main Findings	Highlights
**Ultrasound of diaphragm to diagnose Diaphragm Dysfunction**				
**Lerolle, 2009** [55]	# Cardiac ICU # DE, Pdi, and Gilbert index, MV > 7 days # Severe DD = “Best DE < 25 mm (maximal inspiration)	28	Best DE discriminated patients with and without severe DD	- DE during maxima effort in patients requiring prolonged MV after cardiac surgery may identify patients with severe diaphragmatic dysfunction
**Kim, 2011** [11]	# Medical ICU; MV > 48 h; # During SBT on PSV or T Tube # DD = Diaphragm excursion (DE) < 1 cm	82	- DD prevalence = 29%. - DE < 1.0 cm predicted weaning outcome	DD: - ↑ weaning time and failure - ↑ total ventilation time
**Valette, 2015** [62]	# Medical ICU, # DE during unassisted breathing # DD = DE < 1.0 cm, OR paradoxical OR absent	10	DD in 10 patients	ARF and DD: ↑ high mortality rates (60%)
**Mariani, 2015** [63]	# Medical ICU, MV >7 days and SBT eligible # DE during SBT on T tube # DD = DE < 1.0 cm	34	DD prevalence 38%	DD bilateral in 24% DD: was not associated with extubation failure.
**Lu, 2016** [64]	# Medical ICU, only patients on prolonged MV # TF during SBT on PSV # DD = TF < 20%	41	DD prevalence = 34%.	DD: - Longer ventilation time - Longer ICU stay
**Jung, 2016** [9]	# Patients with ICUAW, MV > 48 h, at 1st SBT # DD = Ptr,stim < 11 cmH_2_O. # Thickening fraction # ICUAW = MRC < 48	40	- DD = 32 patients (80%) - TF < 20% = 70% - TF correlated with Ptr,stim	Patients with ICUAW on first SBT - DD is highly prevalent - TF was able to detect DD
**Dube, 2017** [65]	# MV <24 h after intubation (‘initiation MV’), under ACV # MV at the switch to PSV (‘switch to PSV’) # Ptr,stim and US variables # DD = Ptr,stim <11 cm H_2_O.	112	At the switch to PSV, - TF < 29% reliable to identify DD (sensitivity and specificity of 85% and 88%)	- DD at switch to PSV associated with: - Adverse outcomes (↑ ICU stay, ↑ MV, ↑ mortality)
**Dres M, 2017** [10]	# MV > 24 h, eligible for the 1st SBT # DD = Ptr,stim <11 cm H_2_O # “ICUAW” = MRC < 48. # TF and DE	76	At the first SBT attempt: - DD = 63%; ICUAW = 34%; both = 21%	- ↓ Ptr,stim and TF associated with weaning failure - DD: associated with ↑ ICU and hospital mortality,
**Ultrasound of diaphragm to assess atrophy**				
**Grosu, 2012** [34]	# Daily measurement of Tdi since intubation	7	- Tdi decrease 6% day/MV	- Diaphragmatic thinning occurs since first 48 h of MV
**Schepens, 2015** [35]	# Medical ICU # Daily measurement of Tdi since first 24 h MV	54	- Tdi baseline = 1.9 mm - ± 0.4 mmTdi Nadir = 1.3 mm ± 0.4 mm	- Tdi-exp ⇓ ≈ 32% at the nadir MV - MV duration associated with atrophy
**Goligher, 2015** [23]	# Daily measurement of Tdi since OTI-72h of MV # Inspiratory effort = TF	107	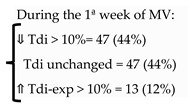	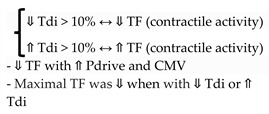
**Zambon, 2016** [36]	# Daily measurement of Tdi since the 1st day of MV # Daily atrophy rate (ΔTdi/d) = % reduction from the previous measurement # MV categorization = 4 classes	40	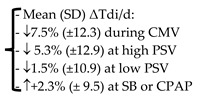	- Only the ventilation support was predictive of diaphragm atrophy rate - Linear relationship between ventilator support and atrophy
**Goligher, 2018** [22]	# Daily measurement of Tdi since intubation until 72h of MV # Inspiratory effort = TF # Primary outcome: time to liberation from MV # 2^ary^ outcomes: complications (reintubation, tracheostomy, prolonged ventilation, or death)	211	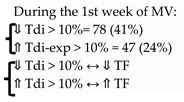	⇓ Tdi > 10%: associated: - Prolonged MV and ICU admission ⇑ Tdi > 10% associated with prolonged MV TF 15–30% = shortest duration of ventilation.
**Dot, 2022** [66]	# Ventilated organ-donors and controls (*n* = 5) # Diaphragm biopsies (to measure CSA) # DUS to measure Tdi (at end-expiration # Median values of the controls = thresholds	35	- All donors presented lower CSA, - 74% donors had lower Tdi - Tdi < 1.7 mm(cut-off) Sens = 73%, Spec = 67%, PPV = 96%;	Diaphragm Tdi: able to detect atrophy but cannot rule out atrophy completely - All donors presented atrophy (decreased CSA)
**Urner, 2022** [67]	# Clinical variables (RR, ventilator settings, and blood gases), were recorded longitudinally. # Diaphragm atrophy (⇓ Tdi > 10% from baseline)	191	- 73 (38%) developed diaphragm atrophy. - TF < 15% during the first 2 d of the study was associated with ⇑ risk of atrophy	- Clinical variables did not predict diaphragmatic contractility or atrophy - The presence or absence of patient-triggered breaths did not influence the risk of diaphragmatic atrophy
**Ultrasound of diaphragm to assess effort**				
**Lerolle, 2009 [55]**	# Cardiac ICU, adult patients > 7 days MV pos-operative # Best DE = DE at maximal inspiratory effort # Pes, Pga, Pdi, # Severe DD = Gilbert index ≤ 0	28	- Pdi,max ⇓ = 27 (96%) patients - Gilbert indexes ≤ 0 = 8 (28%) - Best E < 25 mm correlated with Gilbert index	- Best DE significantly correlated with Gilbert index (assess the diaphragm contribution to respiratory pressure swings during quiet ventilation)
**Vivier,** **2012 [68]**	# ICU, adult # Patients under planned NIV post-extubation # During SB and NIV (PS 5, 10, 15) # Pressure–time product (PTPdi) and USD (Tdi and TF)	12	- TF significantly correlated with PTPdi - PTP(di) and TF both decreased as the level of PS increased.	# TF usefulness - Evaluating diaphragmatic function - Assess diaphragm contribution to respiratory workload
**Goligher 2015 [45]**	# ICU, adult patients requiring MV # TF at peak and end-inspiration (airway occluded and diaphragm relaxed) in 9 controls (at varying lung volumes). # EAdi and Pdi	96	- TFdi and EAdi were positively correlated, - TFdi was lower in patients either on partially AMV or CMV compared to healthy subjects	TF significantly correlated with EAdi and Pdi (validation study)
**Umbrello, 2015 [54]**	# Surgical ICU, adult patients, after major elective surgery eligible for an SBT with PSV # PSV 0, 5 and 15 cmH_2_O). #TF and diaphragmatic excursion # Diaphragm and esophageal PTP (PTPdi and PTPes)	25	- TF and PTP (PTPes and PTPdi) reduced with increasing levels of pressure support - TF significantly correlated with PTPdi and PTPes (r = 0.701 and 0.801).	TF is a reliable indicator of respiratory effort in patients undergoing AMV DE should not be used to quantitatively assess diaphragm contractile activity.
**Dube,** **2017 [65]**	# MV <24 h after intubation (‘initiation MV’), under ACV # MV at the switch to PSV (‘switch to PSV’) # Ptr,stim and US variables were measured in, and compared # DD = Ptr,stim < 11 cm H_2_O.	112	A TFdi < 29% could reliably identify diaphragm dysfunction (sensitivity and specificity of 85% and 88%)	- At switch to PSV, TFdi and EXdi were respectively very strongly and moderately correlated to Ptr,stim, (r = 0.87, *p* < 0.001 and 0.45, *p* = 0.001), - DD was associated with increased duration of ICU stay and MV, and mortality.
**Ultrasound of intercostal muscles**				
**Cala, 1998** **[69]**	# Healthy subjects # US image of the 2nd interspace during tidal breathing, RV, FRC ant TLC # Inter-rib distance, parasternal IC thickness and motion of the midpoint	4	- During inspiration, parasternal IC muscle moves ventrally and straightens - Lung volume influences its shape and motion	- Support an intercostal stabilizing function of the IC parasternal - Suggest that IC parasternal may contribute to the inspiratory fall in pleural pressure
**Wallbridge, 2018 [70]**	# COPD patients # US measurement of thickness and echogenicity of 2nd and 3rd parasternal IC muscles, pectoralis major, quadriceps, and diaphragm thickness # CT-measured lateral IC mass	20	- IC thickness moderately correlated with FEV_1_% predicted and quadriceps thickness - Echogenicity correlated negatively with FEV1% predicted - CT-measured lateral IC mass correlate negatively with parasternal US IC thickness	- US may have biomarker potential for the systemic effects of COPD on muscle as well as local disruption of respiratory mechanics - Changes in muscle quantity and quality reflected spirometric disease severity
**Nakanishi, 2019 [71]**	# MV< 48 h # IC muscle thickness measured with US on days 1, 3, 5 and 7 # Change in IC thickness # The relationship of changes in IC thickness with patient characteristics	80	- IC thickness ↓ in 48 (60%), - IC thickness ↑ 15 (19%) - IC thickness unchanged in 17 (21%)	Decreased IC thickness: - Frequently seen in patients under MV - Associates with adverse outcomes: (prolonged MV and length of ICU stay)
**Yoshida, 2019 [72]**	# Healthy volunteers # IC thickness measured using US at rest and at maximal breathing # Anterior, lateral and posterior portions of the IC spaces	12	- IC thickness ↑ in the 1st, 2nd 3rd 4th and 6th IC spaces of the anterior portions - No significant differences between rest and maximal breathing in the lateral or posterior portions	- US imaging IC muscle would be a useful parameter for assessing the respiratory muscle activity in patients
**Umbrello, 2020 [73]**	# MV patients in PSV # Parasternal IC US (thickness and TF) at 3-levels of PSV (baseline, 25% and 50% reduction) # Diaphragm and esophageal PTP (PTPdi and PTPes) and WOB # DUS	21	- Parasternal IC thickening - May discriminate a low inspiratory - Effort or a higheffort in the presence of DD	- Assessment of extra-diaphragmatic respiratory muscle activity can help to identify conditions of increased inspiratory effort in the presence of DD
**Dres, 2020 [74]**	# Evaluation of the parasternal in healthy volunteers # Responsiveness of parasternal to PSV levels in patients # Comparison of parasternal activity in presence or absence of diaphragm dysfunction # Comparison of parasternal activity in case of success/failure of an SBT	93	- Parasternal IC thickening was responsive to the level of ventilator assistance - Parasternal IC thickening was significantly higher in MV patients with DD - The parasternal IC TF was significantly associated with failure of an SBT	- US of the parasternal IC provides novel information regarding the respiratory capacity load balance
**Dres, 2021 [75]**	# Patients intubated ≥ 48 h # 2 h after extubation # Dyspnea (MRC) # US TF of the parasternal IC and the diaphragm # Limb muscle strength	122	- The median ratio of parasternal IC muscle to diaphragm TF was significantly higher and MRC was lower in patients with extubation failure - The TF of IC and its ratio to diaphragm TF showed the highest AUC for an early prediction of extubation failure	- Extubation failure occurred in 21 (17%) out of patientsRespiratory muscle US 2 h after extubation predict subsequent extubation failure

## Data Availability

Not applicable.

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
