# Peer review of "Diaphragm Ultrasound in Critically Ill Patients on Mechanical Ventilation—Evolving Concepts"

_diagnostics, 2023, doi:10.3390/diagnostics13061116_

Round 1

Reviewer 1 Report

In their descriptive review, the authors report the current knowledge about Diaphragm Ultrasound in the critically ill, particularly focusing in the various aspects related to mechanical ventilation. This comprehensive review analyses the multiple variables of an argument that is having increasing interest among ICU physicians. The paper is well written and I have no major observations.

Author Response

Thanks to the reviewer for the comments. In this article, we sought to present a narrative review on Diaphragm Ultrasound in critically ill patients on mechanical ventilation, focusing on the most current aspects with clinical impact.

Reviewer 2 Report

Minor comments only on form

Suggestions are in added text in the paper (see file)

Author Response

(a)Please add definition of Tdi and TF;

Author’s response: Thanks for the appointment. We added the definitions of Tdi and TF on the legend of the figure.

(b)TYPO ERROR Panel E;

Panel E: Probe positioning on lateral intercostal area (at the zone of apposition, bellow the 9th intercostal space, between 2 ribs). Panel F: Bidimensional view displaying the diaphragm as a hypoechoic structure surrounded by 2 hyperechoic lines (pleural and peritoneal membranes, between the 9-10th ribs); Panel G e H: After positioning the linear probe perpendicular to the costal wall, we employ a small counterclockwise rotation of the probe to observe the diaphragmatic membranes in parallel in two-dimensional mode (corresponding ultrasound image on the right in panel H); Panel I: M-Mode view displaying a normal inspiratory thickening of diaphragm during tidal breathing; Panel J: Graphic illustrative of the thickness and thickening measurement.

(c)Please add legend and abbreviations significance for table;

Author’s response: we added legend and abbreviations significance.

“Table 1: an overview of recent evidence on diaphragmatic and intercostal ultrasound assessment, focusing on clinical applications aimed at promoting diaphragmatic protective mechanical ventilation strategies. ICU – Intensive Care Unit; DE – diaphragmatic excursion; Pdi -  transdiaphragmatic pressure; MV – mechanical ventilation; DD - Diaphragm Dysfunction; SBT - Spontaneous breathing trial; PSV: pressure support ventilation; OR – odds ratio; ARF – acute respiratory failure; TF - thickening fraction; ICUAW - intensive care unit acquired weakness; Ptr,stim: twitch tracheal pressure; ACV - Assist control ventilation; US – ultrasound; MRC: Medical Research council; Tdi - Diaphragm thickness;  OTI –   orotracheal intubation;   Pdrive – driving pressure; CMV – controlled mechanical ventilation; ΔTdi/d - Daily atrophy rate; SD: standard deviation; CPAP: Continuous Positive Airway Pressure; CSA – cross-sectional area; PPV - positive predictive value; DUS – diaphragmatic ultrasound; RR – respiratory rate; Pes - esophageal pressure; Pga – gastric pressure; NIV – noninvasive ventilation; SB - spontaneous breathe; PTPdi - diaphragm pressure–time product; EAdi: diaphragmatic electrical activity; AMV: assisted mechanical ventilation; PTPes - esophageal pressure–time product; EXdi – diaphragmatic excursion; C – intercostal; RV: residual volume; FRC: functional residual capacity; TLC: total lung capacity; IC – intercostal; COPD - chronic obstructive pulmonary disease; FEV1 - Forced Expiratory Volume; CT – Computerized Tomographic; WOB: work of breathing”.

Reviewer 3 Report

In the present narrative review, Pauliane Vieira Santana and coll. summarized well-known and evolving concepts on diaphragm ultrasound in critically ill patients. The topic is interesting and the manuscript complete and well referenced. 

Nevertheless, I have some comments:

Please, conclude the introduction section with a short sentence outlining the aim of the current review and summarizing the structure and content of the manuscript

Figure 1: please describe the acronyms. In particular what is “TF”? If it is thickening fraction, then I am not sure that a decrease in TF means atrophy while an increase in TF indicates hypertrophy, as these are usually inferred from the expiratory thickness, not the inspiratory thickening. A decrease over time of thickening fraction, might also be a consequence of weakness, and it does not imply reduction of diaphragm mass. Please, explain better and avoid oversimplistic interpretations

Figure 2, lower panel. In the left quadrant, a linear ultrasound probe is shown, and immediately at its right side the corresponding ultrasound image is shown. However, the probe is placed perpendicular to the longitudinal axis of the ribs, and if one places the probe in that position, the corresponding image is not the one seen in the figure. Please correct

Page 5: “Although it has been reported as a lower limit for normal diaphragm thickness of around 1.5 mm in healthy individuals (56), in very thin healthy women, thickness values of up to 1.2 cm” I think the authors mean “mm”

Page 5: “Diaphragm thickness (Tdi) may depict atrophy, while the thickening fraction (TF) represents diaphragm inspiratory activity (effort) (57-59).” And page 10 “Assessment of the diaphragm thickening fraction (TF) as a surrogate of contractile activity of the diaphragm (the level of respiratory effort) is a reasonable approach (22, 49)“. This is one of the hottest topics in diaphragm research. Indeed, the literature is far from having found an agreement about whether diaphragm thickening indicates and is related with inspiratory effort. Please, do expand on this point, and consider the studies that found a correlation between effort and thickening (Vivier, Intensive Care Med 2012; Umbrello, Crit Care 2015; Goligher, Intensive Care Med 2015; Umbrello, Br J Anaesth 2020) and those which did not (Oppersma, J Appl Physiol; 2017Poulard, Anesthesiology 2022; Steinberg, Intensive Care Med 2022)

Table 1: please include other studies with particular reference to those investigating diaphragm and other inspiratory muscles such as intercostal (see, for instance, Umbrello, Br J Anaesth 2020; Dres, Anesthesiology 2020)

A paragraph on newer methods of diaphragm imaging (speckle tracking, tissue Doppler imaging and so on) is lacking and should definitely be included along with relevant references. Moreover, in paragraph 4.1 please add the recently published EXODUS Delphi consensus statement (Haaksma, Crit Care 2022)

Author Response

Dear Reviewer. Thanks for your appointments to the manuscript.

We attached a PDF providing a poin-by-point response to the comments and questions.

Round 2

Reviewer 3 Report

I am happy with thi revised version

Author Response

Dear reviewer,

Thanks for your previous relevant comments. They improved our manuscript.